# Histogram of Oriented Gradient-Based Fusion of Features for Human Action Recognition in Action Video Sequences

**DOI:** 10.3390/s20247299

**Published:** 2020-12-18

**Authors:** Chirag I. Patel, Dileep Labana, Sharnil Pandya, Kirit Modi, Hemant Ghayvat, Muhammad Awais

**Affiliations:** 1Computer Science & Engineering, Parul Institute of Technology, Parul University, Vadodara 391760, India; dileepkumar.labana@gujgov.edu.in; 2Symbiosis Centre for Applied Artificial Intelligence and Symbiosis Institute of Technology, Symbiosis International (Deemed) University, Pune 412115, India; sharnil.pandya@sitpune.edu.in; 3Sankalchand Patel College of Engineering, Sankalchand Patel University, Visnagar 384315, India; kjmodi.fet@spu.ac.in; 4Computer Science Department, Faculty of Technology, Linnaeus University, P G Vejdes väg, 351 95 Växjö, Sweden; hemant.ghayvat@lnu.se; 5Center for Intelligent Medical Electronics, Department of Electronic Engineering, School of Information Science and Technology, Fudan University, Shanghai 200433, China; 17110720061@fudan.edu.cn

**Keywords:** moving object detection, surveillance system, support vector machines, histogram of oriented gradient, meta-cognitive neural network (MCNN), multiple kernel learning

## Abstract

Human Action Recognition (HAR) is the classification of an action performed by a human. The goal of this study was to recognize human actions in action video sequences. We present a novel feature descriptor for HAR that involves multiple features and combining them using fusion technique. The major focus of the feature descriptor is to exploits the action dissimilarities. The key contribution of the proposed approach is to built robust features descriptor that can work for underlying video sequences and various classification models. To achieve the objective of the proposed work, HAR has been performed in the following manner. First, moving object detection and segmentation are performed from the background. The features are calculated using the histogram of oriented gradient (HOG) from a segmented moving object. To reduce the feature descriptor size, we take an averaging of the HOG features across non-overlapping video frames. For the frequency domain information we have calculated regional features from the Fourier hog. Moreover, we have also included the velocity and displacement of moving object. Finally, we use fusion technique to combine these features in the proposed work. After a feature descriptor is prepared, it is provided to the classifier. Here, we have used well-known classifiers such as artificial neural networks (ANNs), support vector machine (SVM), multiple kernel learning (MKL), Meta-cognitive Neural Network (McNN), and the late fusion methods. The main objective of the proposed approach is to prepare a robust feature descriptor and to show the diversity of our feature descriptor. Though we are using five different classifiers, our feature descriptor performs relatively well across the various classifiers. The proposed approach is performed and compared with the state-of-the-art methods for action recognition on two publicly available benchmark datasets (KTH and Weizmann) and for cross-validation on the UCF11 dataset, HMDB51 dataset, and UCF101 dataset. Results of the control experiments, such as a change in the SVM classifier and the effects of the second hidden layer in ANN, are also reported. The results demonstrate that the proposed method performs reasonably compared with the majority of existing state-of-the-art methods, including the convolutional neural network-based feature extractors.

## 1. Introduction

In machine vision, automatic understanding of video data (e.g., action recognition) remains a difficult but important challenge. The method of recognizing human actions that occur in a video sequence is defined as human action recognition (HAR). In video understanding, it is difficult to differentiate routine life actions, such as running, jogging, and walking, using an executable script. There has been an increasing interest in HAR over the past decade, and it is still an open field for many researchers. The domain of HAR has developed considerably with significant application in human motion analysis [1,2], identification of familiar people and gender [3], motion capture and animation [4], video editing [5], unusual activity detection [6], video search and indexing (useful for TV production, entertainment, social studies, security) [7], video2text (auto-scripting) [8], video annotation, and video mining [9].

Human action recognition is a challenging multi-class classification problem due to high intra-class variability within a given class. To overcome variability issue, we propose a scheme to design a feature descriptor that is highly invariant to the fluctuations present in the classes. In other words, the proposed feature descriptor fuses various diverse features. In addition, this paper addresses various challenges in HAR, such as variation in the background (outdoor or indoor), recognizing the gender of the action performer, variation in clothes worn, and scale variation. We deal with constrained video sequences that involve moving background and multiple actions in single video sequence.

Our contributions in this paper can be summarized in the following way. First, for moving object detection, we use a novel technique by incorporating the human visual attention model [10] making it background-independent. Therefore, its computational complexity is much lower than the algorithm which updates background at regular interval for moving object detection in the video. Second, we propose the feature description preparation layer, which includes the use of the HOG features with the non-overlapping windowing concept. Moreover, averaging the features reduces the size of the feature descriptor. In addition to the HOG, we also use the object displacement, which is crucial to differentiate the action performed at the same location, i.e., zero displacements (like boxing, hand waving, clapping, etc.) or at various locations, i.e., non-zero displacement (like walking, running, etc.). Furthermore, a velocity feature is used at this stage to further identify the overlapping actions having non-zero displacement (like walking, running, etc.). It is based on the observation that speed variation among such actions exists and incorporation of velocity feature can aid the classification. To consider the spatial context in terms of boundaries and smooth shapes of the human body, regional features from Fourier HOG are employed. Finally, we propose six different models for classification to demonstrate the effectiveness of the proposed features descriptor across different types of classifier families.

The rest of the paper is organized in the following way. Section 2 discusses the existing literature on HAR. Section 3 outlines the motivation for feature fusion and briefly describes the HOG, support vector machines (SVMs), artificial neural networks (ANNs), multiple kernel learning (MKL), and Meta-cognitive Neural Network (McNN). In Section 4, the proposed approach for HAR is described. Section 3 also presents the proposed techniques for fusing features. Section 4 presents and discusses the experimental results. Finally, we conclude the paper in Section 5.

## 2. Existing Methods

In the last two decades, most research on human action recognition is concentrated at two levels: (1) feature extraction and (2) feature classification. One of the feature extraction methods is the Dense trajectories approach [11] that extracts features at multiple scales. In addition, these features are sampled for each frame, and based on the displacement information from dense optical flow field actions are classified. In [12], an extension to Dense trajectories was proposed by replacing the Scale-Invariant Feature Transform (SIFT) feature with the Speeded Up Robust Features (SURF) feature to estimate camera motion.

The advantage of these trajectories representations is that they are robust to fast irregular motions and boundaries of human action. However, this method cannot handle the local motion in any action which involves the important movement of the hand, arm, and leg. Therefore, it is not providing enough information for action discrimination. This particular problem is overcome by using important motion parts using Motion Part Regularization Framework (MPRF) [13]. This framework uses Spatio-temporal grouping of densely extracted trajectories, which have been generated for motion part. Objective function for sparse selection of these trajectory groups has been optimized and learned motion parts are represented by fisher vector. Lan et al. again points out in [14] about the local motion of body parts, which result in small changes of intensity, resulting in low-frequency action information. In feature preparation layer, low-frequency action information is not included; therefore, resultant feature descriptors cannot capture enough detail for action classification. In order to address this problem, the Multi-skIp Feature Staking (MIFS) approach was proposed. This approach considers stacking extracted features using differential operators at various scales, which makes the task of action recognition invariant to speed and range of motion offered by the human subject. Due to consideration of various scales in feature building stage, computation complexity is increased in this approach.

In the traditional way, distinct features are derived for representing any human action. However, Liu et al. [15] proposed a human action recognition system, which extracts spatio-temporal and motion features automatically, and this is accomplished by an evolutionary algorithm such as genetic programming. These features are scale and shift invariant and extract color information as well from optical flow sequences. Finally, classification is performed using SVM but the automatic learning needs training process which is time-consuming. The approach in [16] defined the Fisher vector model based on the spatio-temporal local features. Conventional dictionary learning approaches are not appropriate for Fisher vectors extracted from features; therefore, the authors of Reference [16] proposed Multiple Instance Discriminative Dictionary Learning (MIDDL) methods for human action recognition. Recently, frequency domain representation of the multi-scale trajectories has been proposed [17]. The critical points are extracted from the optical flow field of each frame; later multi-scale trajectories are generated from these points and transformed into frequency domain. This frequency information is combined with other information like motion orientation and shapes at the end. The computational complexity of this method is high due to the consideration of the optical flow. The author [18] proposed the skeleton information as a coordinated non-cyclic diagram that gave the kinematic reliance between the joints and bones in the characteristic human body.

Recently proposed, the Deep Convolution Generate Adversarial Network (DCGANs) [19] bridges the gap between supervised and unsupervised learning. The author proposed a semisupervised framework for action recognition, which uses trained discriminator from GAN model. However, the method evaluates the feature based on the appearance of the human and does not account motion in feature building stage. Representation of action is evaluated in terms of distinct action sketches [20]. Sketch formation has been done using fast edge detection. Later on, the person in each particular frame is detected by R-CNN. Furthermore, ranking and pooling are deployed for designing distinct action sketch. Improved dense trajectories and pooling feature fusion are provided to SVM classifier for action recognition. VideoLSTM, a new recurrent neural network architecture, has been proposed in [21]. This architecture can adaptively fit the requirement of given video. This approach exploits new spatial layout of architecture, motion-based attention for relevant spatio-temporal location, and action localization from videoLSTM. In addition to that, there are several methods proposed over the decades  [22].

## 3. Proposed Framework

The proposed HAR framework is shown in Figure 1 and involves three parts: moving object detection, feature extraction, and action classification.

### 3.1. Moving Object Detection

Moving object detection plays crucial role in many computer vision applications. The process involves the classification of pixels from each frame of a video stream as a background or foreground pixel, and a model representing the background is generated. Then, the background is removed from each frame to enable moving object detection and the process is referred to as background subtraction. Popular background subtraction techniques include frame differencing [23,24], Shadow removal [25], Gaussian mixture model(GMM) [26], and CNN-based background removal [27]. However, an algorithm for moving object detection without any background modeling was presented in [28,29,30], and the detailed procedure is given below.

First, an average filter is applied on video sequence I(m,n,t) of size *X* × *Y* for a particular time *t*.

The moving object detection performance of the method is depicted in Figure 2 and Figure 3 for two different video sequences. The first column shows the snapshot from two different videos. A saliency map is shown in the second column, and a silhouette creation is done using morphological operation and shown in third column. The detected moving objects are shown in the fourth column.
(1)$$I_{avg}=I\left(m,n,t\right)⊗A\left(X,Y\right),$$
where *A* represents the avg filter of mask size *X* × *Y*, and ⊗ represents the convolution between two images. Next, a Gaussian filter is employed on the image,
(2)$$I_{gaussian}=I\left(m,n,t\right)⊗G\left(h,\sigma \right)$$

The Gaussian low-pass filter is represented as *G*. The saliency value calculated at each pixel (m,n) is given as
(3)$$dist\left[I_{gaussian}\left(m,n\right),I_{avg}\left(m,n\right)\right]=\left[I_{gaussian}\left(m,n\right)-I_{avg}\left(m,n\right)\right]$$
(4)$$S\left(m,n\right)=dist[I_{gaussian}\left(m,n\right),I_{avg}\left(m,n\right)]$$

The distance between the respective images is represented as dist. S(m,n) contains the moving object obtained from specific video. In the proposed approach, moving object detection is performed as
(5)FG(m,n)=[|I(m,n,t)−I(m,n,t−1)|>Threshold]
where FG(m,n) defines the moving object from the video sequence *I*. Therefore, the moving object detection is faster and computationally efficient, as the method is background-independent. In other words, the time-consuming process of updating the background at regular intervals is not needed.

### 3.2. Feature Extraction

The procedure for extracting feature descriptors from a segmented object is shown in Figure 4, which represents action in a compact three-dimensional space associated with an object, background scene, and variation that appears in the object over time. After detecting and segmenting moving objects from each video sequence, compact features are extracted. In the proposed approach, we calculate the following features.
HOG over 10 non-overlapping frames (HOGAVG10):Here, we have used HOG, which was proposed by Dalal and Trigg [31] in 2005 and is still a highly effective human detection feature. The segmented object is converted to a fixed size (e.g., 128 × 64). HOG features extracted from the resized segmented object (per frame) have a dimensionality of 3780 as explained in Figure 4. Each video has 120 frames; therefore, the final descriptor for each video having one action is 3780 × 120. Feature descriptors contain redundant data; thus, the computational cost for learning and testing is excessive. In the proposed approach, we have calculated HOG features over a window size of 10 non-overlapping frames (HOGAVG10) because the object does not change considerably over the frames as shown in Figure 5 . Thus, there is a considerable reduction in the redundant data by using 10 frames.Displacement in Object Position (OBJ_DISP):To evaluate the displacement of an object, the centroid (or center of mass) of the silhouette corresponding to the object is calculated by taking the (arithmetic) mean of the pixels is denoted by
(6)$$\mu \left(x_{i},y_{j},t\right)=\frac{1}{nm}\sum_{i=1}^{n}\sum_{j=1}^{m}C(x_{i},y_{j})$$Suppose that the centroid of the present frame is $C(x_{t},y_{t},t)$ and the past frame is $C(x_{t-1}$, $y_{t-1},\;t-1)$. Then, the displacement (OBJ_DISP) $D(x_{t},y_{t},t)$ can be approximated using
(7)$$OBJ\_DISP\left(x_{t},y_{t},t\right)=\sqrt{{\left(x_{t}-x_{t-1}\right)}^{2}+{\left(y_{t}-y_{t-1}\right)}^{2}}$$Velocity of Object (OBJ_VELO):Similar to the displacement features, the extraction of velocity features also requires the centroid of the detected moving object. The displacement and velocity features are used to estimate the motion of the moving object as they increase the inter-class distance which subsequently increases the accuracy of the overall proposed framework.Velocity OBJ_VELO$\left(x_{t},y_{t},t\right)$ of object is estimated using
(8)$$OBJ\_VELO\left(x_{t},y_{t},t\right)=\frac{OBJ\_DISP(x_{t},y_{t},t)}{\Delta t},$$
where Δ*t* = $t_{i+10}-t_{i}$ (for example, Δt=10 for our proposed approach) and OBJ_DISP refers to Displacement.Regional Features from Fourier HOG [32] (R_FHOG):In this work, we extended the Regional Features from Fourier HOG proposed in [32] for action recognition. In Cartesian coordinate system, two-dimensional function is represented by $f\left(x,y\right)\in R^{2}$. The polar coordinate representation of same function is defined as [r,θ], as *r* is frequency in radius and angle θ. The relation between polar and Cartesian coordinate is defined as
(9)$$r=\left\|f\right\|=\sqrt{x^{2}+y^{2}},$$
and
(10)θ=arctan(y,x)∈[0,2π).In the polar coordinate system, the Fourier transform is combination of radial and angular parts. The basis function B for Fourier transform in polar coordinate systems is defined as
(11)$$B_{k,m}\left(r,\theta \right)=\sqrt{k}J_{m}\left(kr\right)\Phi_{m}\left(\theta \right),$$
where *k* is non-negative value, and its also defines the scale of the pattern; $J_{m}\left(kr\right)$ is a *m*th-order Bessel function; and $\Phi_{m}=\frac{1}{2}e^{im\Phi }$. *k* can be continuous or discrete value, depending on whether the region is infinite or finite. Transform considering finite region r≤a, the basis function is reduced to
(12)$$B_{n,m}\left(r,\theta \right)=\sqrt{k}R_{nm}\left(r\right)\Phi_{m}\left(\theta \right),$$
where,
(13)$$R_{nm}\left(r\right)=\frac{1}{\sqrt{N_{n}^{\left(m\right)}}}J_{m}\left(k_{nm}r\right).$$

The basis function (Equation 13) is orthogonal and orthonormal in nature. For $B_{n,m}\left(r,\theta \right)$, *m* is number of cycles in angular direction and n−1 is defined as number of zero crossing in radial direction.

As the values of m and n increase, finer details can be extracted from the image. Generally, the evaluation of HOG features involves three steps namely gradient orientation binning, spatial aggregation, and magnitude normalization, which are followed in the Fourier domain as well. *Step 1:* Gradient Orientation Binning:

The gradient of image $I\left(x,y\right)\in R^{2}$ is defined as $G\left(x,y\right)=\left[G_{x},G_{y}\right]$, and its polar representation is defined as
(14)$$F_{m}\left(r,\theta \right)=∥G∥e^{-i∠G},$$
where $∥G∥=\sqrt{G_{x}^{2}+G_{y}^{2}}$ and $∠G=\arctan\left(G_{y},G_{x}\right)\;\in \left[0,2\pi \right)$. Gradient orientation are stored in bins of histogram using distribution function h at each pixel. Suppose that the gradient of any image is resented as $G=\left[G_{x}G_{y}\right]\in R^{2}$. The angular part of *G* is Φ(G), and the distribution function h for each pixel should be Dirac function gain with ‖G‖
(15)h(θ)=‖G‖δ(θ−Φ(G)).

In this work, Fourier basis has been replaced with Fourier coefficient ${\hat{f}}_{m}$
(16)$${\hat{f}}_{m}=\langle h,e^{im\phi }\rangle =∥G∥e^{-im\phi (G)}.$$

In HOG, for each gradient vector, its magnitude contribution is split into three closest bins. Therefore, it can be considered a triangular interpolation. In Fourier space, to build a HOG feature, a 1D triangular kernel can be employed to implement the gradient orientation binning. However, the execution of this particular step does not affect the results. Therefore, this step has not been considered in the proposed work. *Step 2:* Spatial Aggregation:


To achieve spatial aggregation, convolution operation is performed on a Gaussian Kernel or an isotropic kernel and Fourier coefficients obtained.*Step 3:* Local Normalization:

An isotropic kernel is convolved with Fourier coefficient to achieve normalization of gradient magnitude. Steps 2 and 3 are performed using two kernels. The first kernel for spatial aggregation is $K_{1}:R^{2}\to R$ and the second kernel $K_{2}:R^{2}\to R$ is used for local normalization. Finally, Fourier HOG is accomplished using
(17)$${\tilde{F}}_{m}=\frac{F_{m}*K_{1}}{\sqrt{{\left\|G\right\|}^{2}*K_{2}}}.$$
Regional descriptor using Fourier HOG:

To obtain the regional descriptor, a convolution operation is performed using the Fourier basis (in polar representation) function. $B_{n,m}\left(r,\theta \right)$.
(18)$$R_{n,m}=B_{n,m}\left(r,\theta \right)*{\tilde{F}}_{m}.$$

The graphical illustration of calculation of R_FHOG features is provided in Figure 6. Figure 7 depicts the positive result by showing R_FHOG (i.e., $R_{n,m}$) for the segmented object. To speed up the process, we have not considered non-redundant data. Therefore, we have selected region features which give a maximum response on a human region. The formation of the final template from region features considers a value of scale ∈{1}, order ∈{1,−1}, and degree ∈{1,2}. Template has been shown in Figure 8.

### 3.3. Fusion of Features

The motivation behind fusing features is to increase diversity within classes and thus improve classification.
HOGAVG10 + OBJ_DISP:Here, we fuse HOGAVG10 with OBJ_DISP. The importance of this parameter is to differentiate between actions performed at a static location (e.g., boxing, hand waving, and hand clapping) and actions performed at a dynamic location (e.g., walking, jogging, and running). Therefore, we gain inter-class discriminative power by combining these two features.The position of an object does not change drastically; thus, we propose to employ the window concept to investigate the object motion over that period. In addition, we take the average of the positions to reduce the feature set. This feature is important as it provides the inter-frame offset corresponding to the object position. The displacement values for all classes are shown in Table 1.HOGAVG10 + OBJ_VELO:Actions with smaller interclass distances such as walking, jogging and running can be distinguished using velocity features. Therefore, we propose to fuse HOGAVG10 with OBJ_VELO.HOGAVG10 + OBJ_DISP + OBJ_VELO:The HOGAVG10 + OBJ_DISP feature combination can differentiate actions performed at static/dynamic locations, whereas the HOGAVG10 + OBJ_VELO feature combination can effectively differentiate classes with similar actions. Therefore, we propose to combine HOGAVG10 + OBJ_DISP + OBJ_VELO to effectively classify similar actions performed at static/dynamic locations present in KTH and Weizmann datasets. The velocity values of persons performing actions are reported in Table 2.R_FHOG + HOGAVG10 + OBJ_DISP+ OBJ_VELO:The R_FHOG feature is effective at splitting the frequency gradient into bands, subsequently emphasizing the human action region. In other words, R_FHOG represents crucial information regarding boundaries and smoothed shapes. R_FHOG also provides information regarding the spatial context of a human subject.

### 3.4. Formal Description

This section presents the proposed fusion techniques in detail. Fusion techniques are performed at both feature and classifier level, referred to as early and late fusion techniques, respectively.

#### 3.4.1. Early Fusion

The task Feature Fusion is performed using basic techniques such as concatenating features one after another as shown in Figure 9.

#### 3.4.2. Late Fusion

Late combination is utilized in this work to accomplish combination at classifier level. The two distinctive late combination approaches utilized in the current investigation are Decision Combination Neural Network (DCNN) and Sugeno Fuzzy Integral.
**Decision Combination Neural Network (DCNN)** 

Decision Combination of Neural Network (DCNN) [33] is neural network architecture with no concealed layers. Accordingly, DCNN characterizes the straight connection between the input and output nodes. The most elevated reaction of a specific output layer node is characterized as choice or class mark for action recognition. Details of the DCNN follow.

As shown in Figure 10, this neural network organization contains two layers: input layer (*S*) and output layer (*Z*) individually. *M* classifier’s outputs are taken care of corresponding to the input layer and there are *N* inputs nodes related with the class. The connection between units (/nodes) of the input layer and output layer are between associated by weights *w*. Each input node gets a score $s_{ik}$, where *i* characterizes *i*th classifier and *k* characterizes *k*th class. On the off chance that $s_{ik}$ of input is associated with output node *j*, the weight of this connection is characterized as $w_{ijk}$. The greatest reaction at the output layer node is characterized as the choice of action recognition.

The sigmoid activation function is used in each node, the reaction of this proposed late combination approach is characterized as
(19)$$h_{j}\left(S_{1},\ldots ,S_{M}\right)=\sum_{i=1}^{M}\sum_{j=1}^{N}w_{ijk}s_{ik},$$
(20)$$DCNN\left(S_{1},\ldots ,S_{M}\right)=\frac{1}{1+e^{-h_{j}\left(S_{1},\ldots ,S_{M}\right)}}.$$
**Sugeno’s Fuzzy Integral** 

The supposition of a basic weighted normal system is that all classifiers are not commonly reliant. In any case, classifiers are connected. To take out a requirement for such presumption, the possibility of fuzzy integral was actualized by the authors of [34,35] and is a nonlinear mapping function characterized with a fuzzy measure. A fuzzy integral is the fuzzy normal of classifier scores. Definitions are given underneath thinking about fuzzy and fuzzy integral, separately.

**Definition** **1.**
*Let X be a finite set defined as $\left\{x_{1},x_{2}\ldots x_{n}\right\}$. A fuzzy measure μ defined on X is a set of function μ:X→[0,1] satisfying with*
*1.*   
*μ(ϕ)=0, μ(X)=1,**2.*   
*A⊆B, μ(A)≤μ(B).*

*The fuzzy measure we adopt in this work is the Sugeno integral.*


**Definition** **2.**
*Let μ be a fuzzy measure on X. The discrete Sugeno integral of function f:X→[0,1] with respect to μ is defined as*
(21)$$S_{\mu }\left(f\left(x_{1}\right),f\left(x_{2}\right)\ldots f\left(x_{n}\right)\right)≜\vee_{i=1}^{n}(f\left(x_{i}\right)\wedge \mu \left(A_{\left(i\right)}\right)$$
*where, (i) shows the indices have been permuted so that $0\leq f\left(x_{1}\right)\leq f\left(x_{2}\right)\ldots f\left(x_{n}\right)\leq 1$. Moreover, $A_{\left(i\right)}:=\left\{x_{\left(i\right)}\ldots x_{\left(n\right)}\right\}$ and $f(x_{\left(0\right)})=0$.*


Fuzzy measure μ is a $\mu_{\lambda }$-fuzzy measure and is calculated by using Sugeno’s λ measure. The value of $\mu (A_{\left(i\right)})$ is calculated recursively as
(22)$$\mu \left(A_{\left(1\right)}\right)=\mu \left(\left\{x_{\left(1\right)}\right\}\right)=\mu^{1}$$
(23)$$\mu \left(A_{\left(i\right)}\right)=\mu^{i}+\mu \left(A_{\left(i-1\right)}\right)+\lambda \mu^{i}\mu \left(A_{\left(i-1\right)}\right)\;for\;1<i\leq n.$$
the value of λ is calculated by solving the equation
(24)$$\lambda +1=\prod_{i=1}^{n}\left(1+\lambda \mu^{i}\right)$$
where λ∈(−1,+∞) and λ≠0. This can be easily computed by calculating an (n−1)st degree polynomial and determining the distinct root greater than −1. The fuzzy integral is characterized in proposed work as late combination method for consolidating classifiers scores. Assume that $C=\{c_{1},c_{2}\cdots c_{n}\}$ is a bunch of action classes of interest. Let $X=\{x_{1},x_{2}\ldots x_{n}\}$ be a bunch of classifiers and *A* be an input pattern considered for action recognition. Let $f_{k}:X\to \left[0,1\right]$ be the assessment of the object *A* for class $c_{k}$, for example, $f_{k}\left(x_{i}\right)$ is sign of guarantee in the characterization of the input pattern *A* for class $c_{k}$ utilizing the classifier $x_{i}$. Value 1 for $f_{k}\left(x_{i}\right)$ is characterizing outright guarantee of input pattern *A* in class $c_{k}$ and 0 shows supreme uncertainty that the object is in $c_{k}$.

Knowledge of the density function is needed to figure the fuzzy integral and $\mu^{i}$, *i*th density is considered as the level of significance of the source $x_{i}$ towards a ultimate choice. A maximal evaluation of comprehension between the evidence and desire is spoken to as fuzzy integral. In the proposed approach, the density function μ is approximated via preparing information gave to the classifier. The calculation in the proposed algorithm characterizes the late combination approach for choice combination. The Algorithm 1 defines the late fusion approach for decision fusion.
**Algorithm 1** Late fusion (decision fusion) using fuzzy integral.**procedure**Fuzzy –Integral    Calculate λ;    **for** each action class $c_{k}$
**do**        **for** each classifier $x_{i}$
**do**           Compute $f_{k}\left(x_{i}\right)$           Determine $\mu_{k}\left(\left\{x_{i}\right\}\right)$        **end for**        Calculate fuzzy integral for the action class    **end for**    Find out the action class label**end procedure**

### 3.5. Classifier

Various classifiers have been used to evaluate the performance of proposed approach. The parameters and their respective values are summarized in Table 3. We have considered the parameters kernel function with degree (d), Gamma in Kernel Function (γ), and Regularization Parameter (c). Polynomial and Radial basis kernel functions have been used.

The parameters of the ANN are hidden layer neurons (n), the value of the learning rate (lr), momentum constant (mc), and number of epochs (ep). To find out the values of these parameters efficiently, ten levels of n, nine levels of mc, and ten levels of ep are evaluated in the parameter setting experiments. The value of lr is initially fixed at 0.1. The values of these parameters and their respective levels are evaluated in Table 4.

#### 3.5.1. Meta-Cognitive Neural Network (McNN) Classifier

Neural network provides a self-learning mechanism, whereas the meta-cognitive phenomenon comprises self-regulated learning. Self-regulation makes the learning process more effective. Therefore, there is need of jump from single or simple learning to collaborative learning. The collaborative learning can be achieved using the cognitive component, which interprets knowledge, and the meta-cognitive component, which represents the dynamic model of the cognitive component.

Self-regulated learning is a key factor of meta-cognition. It is threefold mechanism: it plans, monitors, and manages the feedback. According to Flavell [37], meta-cognition is awareness and knowledge of the mental process for monitoring, regulate, and direct the desired goal. We present here Nelson and Naren’s meta-cognitive model [38]. The cognitive component and meta-cognitive component are prime entities of McNN. A detailed architecture of the Meta-cognitive Neural network is shown in Figure 11.

#### 3.5.2. Cognitive Component

The cognitive component includes three-layered feedforward radial basis function network. It comprises an input layer, an output layer, and an intermediate hidden layer. The activation function for hidden neurons is Gaussian whereas, for output neurons, it is a linear activation function. Hidden layer neurons are built by the meta-cognitive algorithm. The predicted output $\hat{y}$ of the McNN classifier with *k* Gaussian neurons from i−1 training samples is
(25)$${\hat{y}}_{j}^{i}=\alpha_{j0}+\sum_{k=1}^{K}\alpha_{jk}\phi_{k}\left(x^{i}\right),\;j=1,2,\ldots ,n,$$
where $\alpha_{j0}$ = bias to *j*th output neuron, $\alpha_{jk}$ is weight connecting the *k*th neuron to the *j*th output neuron, and $\phi_{k}\left(x^{i}\right)$ is the output of *k*th Gaussian neuron to the excitation *x* is represented as
(26)$$\phi_{k}\left(x^{i}\right)=exp\left(-\frac{{∥x^{i}-\mu_{k}^{l}∥}^{2}}{{\left(\sigma_{k}^{l}\right)}^{2}}\right),$$
where $\mu_{k}^{l}$ is the mean, $\sigma_{k}^{l}$ is the variation in the mean value of the *k*th hidden neuron, and *l* represents the hidden layer class.

#### 3.5.3. Meta-Cognitive Component

*Measures:* 

The meta cognitive component of McNN uses four parameters for regulation learning:Estimated class label:Estimated class label $\hat{c}$ can be calculated from predicted output ${\hat{y}}^{i}$ as
(27)$$\hat{c}=arg\;{max}_{j\in 1,2,\ldots n}\;{\hat{y}}_{j}^{i}.$$Maximum hinge error:Hinge error estimates posterior probability more precisely than mean square error function, and, eventually, the error between the predicted output ${\hat{y}}^{i}$ and actual output $y^{i}$ hinge error loss defined as
(28)$$e_{j}=\left\{\begin{array}{cc} 0 & if\;{\hat{y}}_{j}^{i}y_{j}^{i}>1\\ {\hat{y}}_{j}^{i}-y_{j}^{i} & otherwise, \end{array}\right.j=1,2,\ldots ,n$$The maximum absolute hinge error *E* is as follows,
(29)$$E={max}_{j\in 1,2,\ldots n}\;\left|e_{j}\right|.$$Confidence of classifier:The classifier confidence is given as
(30)$$\hat{p}\left(c/x^{i}\right)=\frac{min(1,max\left(-1,{\hat{y}}_{j}^{i}\right))+1}{2}.$$Class-wise significance:The input feature is mapped to higher dimensional S using Gaussian activation function applied to hidden layer neurons. Therefore, it is considered to be on hyper-dimensional sphere. The feature space S is described by the mean μ and σ variation in the mean value of Gaussian neurons. Moreover, steps are shown in [39] for the calculation of potential ψ, which is given as
(31)$$\psi \approx -\frac{2}{K}\sum_{k=1}^{K}\phi \left(x^{i},\mu_{k}^{l}\right).$$In the classification problem, each class distribution is considered crucial and eventually affect the accuracy of the classifier, significantly. Therefore, a measure of the spherical potential of new training data x belongs to class c with respect to neurons belongs to same class has been utilized, i.e., l=c. Class-wise significance $\psi_{c}$ is calculated as
(32)$$\psi_{c}=\frac{1}{K^{c}}\sum_{k=1}^{K^{c}}\phi \left(x^{i},\mu_{k}^{c}\right),$$
where $K^{c}$ is the number of neurons associated with class *c*. The sample contains relevant information or not depends on $\psi_{c}$, the lowest value of it denotes sample consider novelty.

*Learning Strategy:* 

Based on various measures, the meta-cognitive component has different learning strategies, which deal with the basic rules of self-regulated learning. These strategies manage sequential learning process by utilizing one of them for new training sample.

Sample Delete Strategy:This strategy reduces the computational time consumed by learning the process. It reduces the redundancy in training samples, i.e., it prevents similar samples being learnt by the cognitive component. The measures used for this strategy are predicted class label and confidence level. When actual class label and predicted class label of the new training data is equal and the confidence score is greater than expected value, it indicates that new training data training data provides redundancy.Neuron growth strategy:New hidden neuron should be added to the cognitive component or not is decided by this strategy. When new training sample include substantial information and the estimated class label is different from an actual class label, new hidden neuron should be added to adopt the knowledge.Parameter update strategy:Parameters of the cognitive component are updated in this strategy from new training sample. The value of parameters change when an actual class label is same as the predicted class of sample and maximum hinge loss error is greater than a threshold set for adaptive parameter updation.Sample reverse strategy:Fine tuning of parameters of the cognitive component has been established by new training samples, which are having some information but not much relevant.

The parameters are updated in McNN, when the desired class is equal to the actual class. The value of maximum hinge error E for neuron growth and the parameter update strategy is between 1.2 and 1.5, and 0.3 and 0.8, respectively. For parameter update strategy, If the value is close to 1, it will avoid system to use any sample. The value is close to 0 cause all samples to be used in updation. In neuron addition strategy, the value 1 lead of E lead to misclassification of all samples and the value 2 causes few neurons will be added. Other parameters are selected accordingly and the range of values of parameters have been shown in Table 5.

## 4. Performance Evaluation

A performance evaluation of the proposed work has been done using a sufficient set of performance parameters through extensive experiments on standard datasets, which is described as follows.

### 4.1. Database Used

The proposed approach was applied to two datasets: the KTH [40] and Weizmann datasets [41]. These datasets are popular benchmarks for action recognition in constrained video sequences. These datasets incorporate only one action in each frame with the static background.

#### 4.1.1. KTH Dataset

The KTH dataset contains action clips with variations in the background, object, and scale, and was thus useful for determining the accuracy of our proposed method. The video sequences contain six different types of human actions (i.e., walking, jogging, running, boxing, hand waving, and hand clapping) performed several times by 25 subjects in four different scenarios: outdoors, outdoors with scale variation (zooming), outdoors with different clothes (appearance), and indoors, as illustrated below. Static and homogeneous backgrounds are considered in all sequences, where the frame rate is 25 frames per second. The resolution of these videos is 160 × 120 pixels, and the duration of the videos is four seconds on average. There are 25 videos for each action in the four different categories. Certain snapshots of video sequences from the KTH dataset are shown in Figure 12.

#### 4.1.2. Weizmann Dataset

The Weizmann database [41] is a collection of 90 low-resolution (180 × 144, de-interlaced 50 frames per second) video sequences. The dataset contains nine different humans, each one performing ten natural actions: run, walk, skip, jumping-jack (or shortly jack), jump forward on two legs (or jump), jump in place on two legs (or pjump), gallop sideways (or side), wave two hands (or wave2), wave one hand (or wave1), or bend. Snapshots of the Weizmann dataset are shown in Figure 13.

#### 4.1.3. UCF11 Dataset

The UCF11 dataset [42] considered 11 human action with 1600 videos. These videos comprise youtube videos defining real human actions. The actions are performed by 25 various human objects under challenging conditions like large changes in viewpoint change, object scale, object appearance and pose, camera motion, cluttered background, illumination variation, etc. There are 11 action categories in UCF11: basketball shooting (*Shoot*), biking/cycling (*Bike*), diving (*Dive*), golf swinging (*Golf*), horse back riding (*Ride*), soccer juggling (*Juggle*), swinging (*Swing*), tennis swinging (*Tennis*), trampoline jumping (*Jump*), volleyball spiking (*Spike*), and walking with a dog (*Dog*).

#### 4.1.4. HMDB51 Dataset

The HMDB51 dataset [43] is built up using videos, adopted from YouTube, movies, and various other sources for managing unconstrained environment. The datasets have the variety of 6849 video clips and 51 action categories. Each class has the at least 101 clips.

#### 4.1.5. UCF101 Dataset

UCF101 is a dataset [44] of 13,320 videos including 101 different action classes. This dataset reflects the large diversity in terms of human appearance performing the action, scale, and viewpoint of the object, background clutter, and illumination variation, resulting in the most challenging dataset. This dataset is bridging path to real-time action recognition.

### 4.2. The Testing Strategy

The KTH dataset contains 600 video samples of 6 types of human actions. The dataset is divided into two parts: 80% and 20%. We have used a 10-fold leave-one-out cross-validation scheme on the 80% part and left out 20% for testing. In this experiment, nine splits are used for training, with the remaining split being used for the validation set, which optimizes the parameters of each classifier. The same testing strategy has been implemented for Weizmann dataset. Leave-One-Group-Out cross-validation has been used for the UCF11 dataset. A cross-validation strategy used for the HMDB51 dataset, the same as in [43]. The whole dataset is divided into three portions. Each includes 70 training and 30 testing video clips. The training strategy used for UCF101 is three split technique evaluated for training and testing.

### 4.3. Experimental Setup

Experiments were performed on an Intel(R) Core(TM) i5 (2nd Gen) 2430M CPU @ 2.5 GHz with 6 GB of RAM and a 64-bit operating system. The names of the parameters and the values used in this proposed work are listed in Table 3 and Table 4, respectively. In this section, we examine the performance of our proposed approach and compare it with the state-of-the-art methods. We also compare different classifier performances with our feature extraction technique for the proposed framework. All confusion matrices address the average accuracy of all features for the SVM classifier with different kernel functions, as well as for the ANN with different numbers of hidden layers.

In this experiment, we have also considered different types of fusion techniques, i.e., early and late have been considered for experimentation. We have employed five various fusion strategies in the proposed work. Figure 14, Figure 15, Figure 16, Figure 17, Figure 18 and Figure 19 present the various models of the early fusion and late fusion techniques used in our experiments. In Figure 14, early fusion has been applied to features and fed to ANN classifier, and some early fusion of features are fed to SVM classifier as shown in Figure 15. Features are provided to MKL with base learner as ANN, and MKL with base learner as SVM, these strategies are defined in Figure 16 and Figure 17, respectively. Figure 18 shows a combination of classifiers scores using late fusion techniques, where we have used SVM classifier in this technique. Meta-cognitive Neural network has been used with all proposed features as shown in Figure 19.

### 4.4. Empirical Analysis

The confusion matrix is shown in Table 6 and Table 7 for different combinations of feature extraction and classifier techniques for the KTH dataset. Table 8, Table 9 and Table 10 show the results with the Weizmann dataset. We have considered linear, polynomial, and radial basis kernel functions for the SVM classification. The results demonstrate that we obtain the good result (97%) with the radial basis function SVM and best result 99.98% with the late fusion using fuzzy integral approach compare to other proposed approaches. Ambiguity arises from the classes like boxing, hand waving and hand clapping actions. Furthermore, running, walking and jogging are misclassified by all classifiers.

The confusion matrix with Radial basis function SVM (RBF SVM) for the UCF11 dataset is shown in Table 11. We accomplished 77.05% accuracy in this proposed approach for UCF11 dataset with previously mentioned parameters SVM classifier as in Table 4. Table 12, Table 13 and Table 14 shows the confusion matrix for KTH, Weizmann & UCF11 dataset using McNN. The UCF11 dataset has unconstrained environments and contains various challenges in video sequences; the proposed feature extraction technique is not adequate for describing the action performed by the human object. Therefore, we can see that a lot of actions are misclassified into other actions like *Shoot* is misclassified as *Swing*, etc.

Accuracy obtained for KTH dataset using late fusion using DCNN is 99.19% and late fusion using fuzzy integral is 99.98% i.e., effectiveness of fuzzy integral technique compared to DCNN technique as late fusion is higher as shown in Table 15.

Moreover, the performance of five broad groups is evaluated in this work using the particular model as shown in Figure 20. Recognition rate has been calculated for all group categories. A Large portion of performance has been gain from sports category. Even all other categories are performing impressively.

Table 15 and Table 16 compare our results with the state-of-the-art methods. Table 15 compares our proposed approach with 21 other approaches that used the KTH dataset. Our approach obtained an accuracy of 100%, which is outperformed to those of the state-of-the-art methods. The proposed approach is compared with the state-of-the-art methods for the Weizmann dataset, which is shown in Table 16. The result shows that our method outperforms the other methods. These comparisons demonstrate that the proposed approach is effective and superior in classifying actions.

Table 17, Table 18 and Table 19 show the state-of-the-art comparison for UCF11 dataset, HMDB-51 dataset and UCF101 dataset, respectively. Our results are achieving very good classification rate compared to other approaches, but humbler than the state-of-the-art results. Compare to early fusion and intermediate fusion techniques, late fusion techniques are superior. In late fusion techniques, fuzzy integral is performing better than DCNN late fusion technique for UCF11 dataset.

In Table 20, we compare our approach with various convolutional neural network architectures. For this comparison, the average accuracy has been calculated over three splits as is the original setting. For the UCF101 dataset, we find that our McNN with proposed features performed well compared with state-of the-art methods. For UCF101, we get a 1% improvement in classification accuracy. However, our result for HMDB51 dataset is not the best result, but the improvement in resultant accuracy is considerable.

## 5. Conclusions

In this paper, we have employed a HAR-based novel feature fusion approach. HOG, R_FHOG, displacement, and velocity features are combined to prepare the feature descriptor in this approach. The classifiers used to classify human action are an ANN, a SVM, MKL, late fusion approach, and McNN. The experimental results demonstrate that this proposed approach can easily recognize actions such as running, walking, and jumping. The McNN outperforms other classifiers. The proposed approach performs reasonably well compared with the majority of existing state-of-the-art methods. For the KTH dataset, our proposed approach outperforms existing methods, and for the Weizmann dataset our approach performs similarly to standard available methods. We have also checked the system performance with unconstrained UCF11 dataset, HMDB51 dataset, and UCF101 dataset, and its performance is approximate to the state-of-the-art method.

In the future, an overlapping window can be utilized for the feature extraction technique to increase the accuracy of the proposed method. Here, the proposed work focuses only on a constrained video; however, we can also use this proposed feature set for an unconstrained video, where more than one object is present in the video performing the same action or in the video performing multiple actions. The traditional neural network can be replaced by the convolutional neural network for further enhancements. We can conclude that fusion of features is a vital idea to enhance the performance of the classifier, where a large complex set of features available. Late fusion was found to be better than early fusion as features are used by multiple classifiers because of their competitiveness for late fusion.

## Figures and Tables

**Figure 1 sensors-20-07299-f001:**
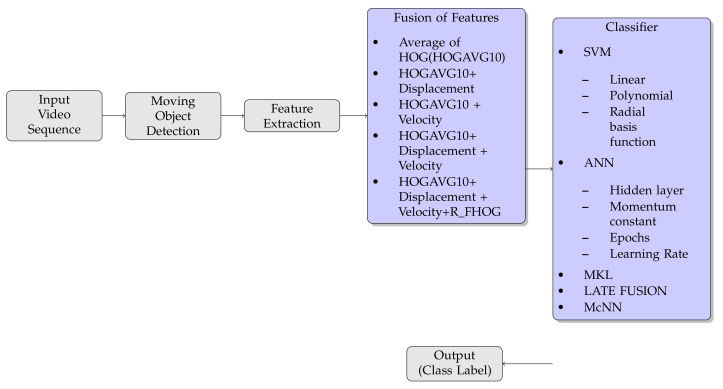
Proposed framework.

**Figure 2 sensors-20-07299-f002:**
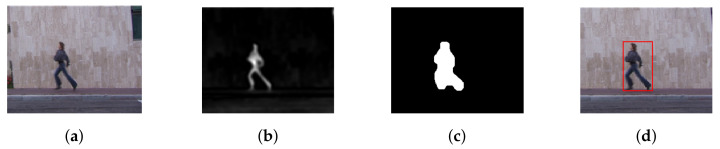
Moving Object Detection: (**a**) Video Sequence. (**b**) Saliency Map. (**c**) Silhouette Creation. (**d**) Segmented Object Image.

**Figure 3 sensors-20-07299-f003:**
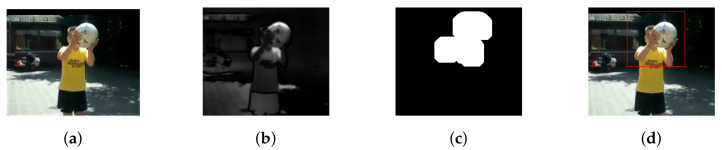
Moving Object Detection: (**a**) Video Sequence. (**b**) Saliency Map. (**c**) Silhouette Creation. (**d**) Segmented Object Image.

**Figure 4 sensors-20-07299-f004:**
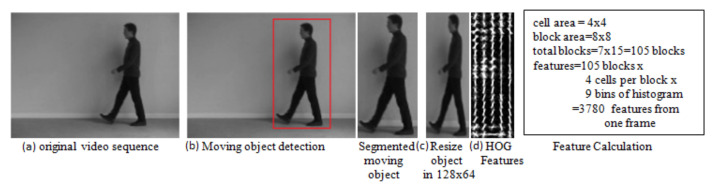
Proposed feature extraction technique: (**a**) Original video sequence. (**b**) Detected moving object. (**c**) Resize detected moving object into 128 × 64 size. (**d**) Histogram oriented gradient (HOG) feature extraction.

**Figure 5 sensors-20-07299-f005:**
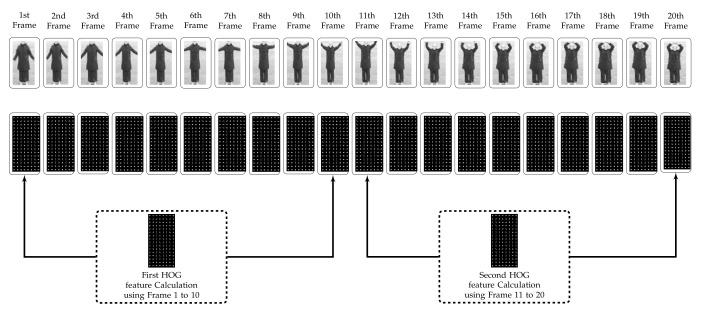
Proposed feature calculation scheme.

**Figure 6 sensors-20-07299-f006:**
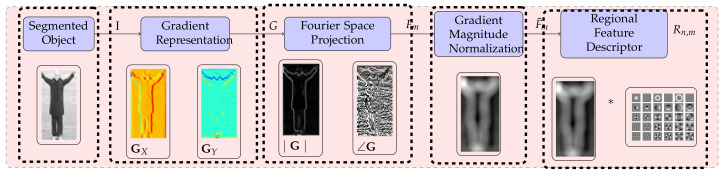
The generation process of the Region Feature Description.

**Figure 7 sensors-20-07299-f007:**
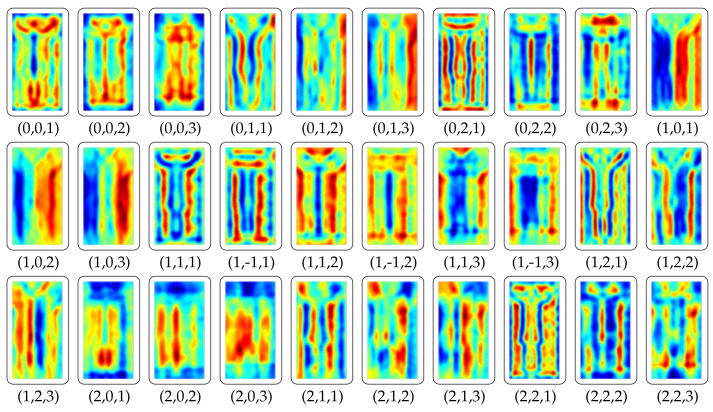
The generation process of the Region Feature Descriptor for segmented moving object: The value below each descriptor image is defined as the scale (*k*), order (*m*), and degree (*n*) of the Basis function $B_{n,m}\left(r,\theta \right)$.

**Figure 8 sensors-20-07299-f008:**
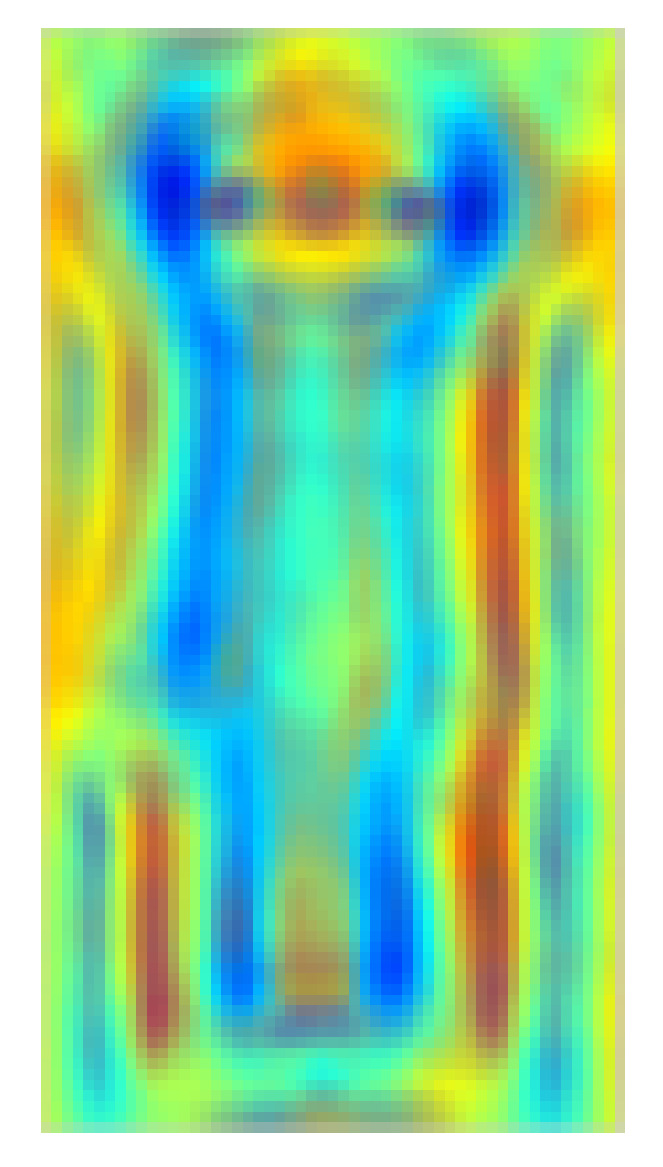
R_FHOG template.

**Figure 9 sensors-20-07299-f009:**
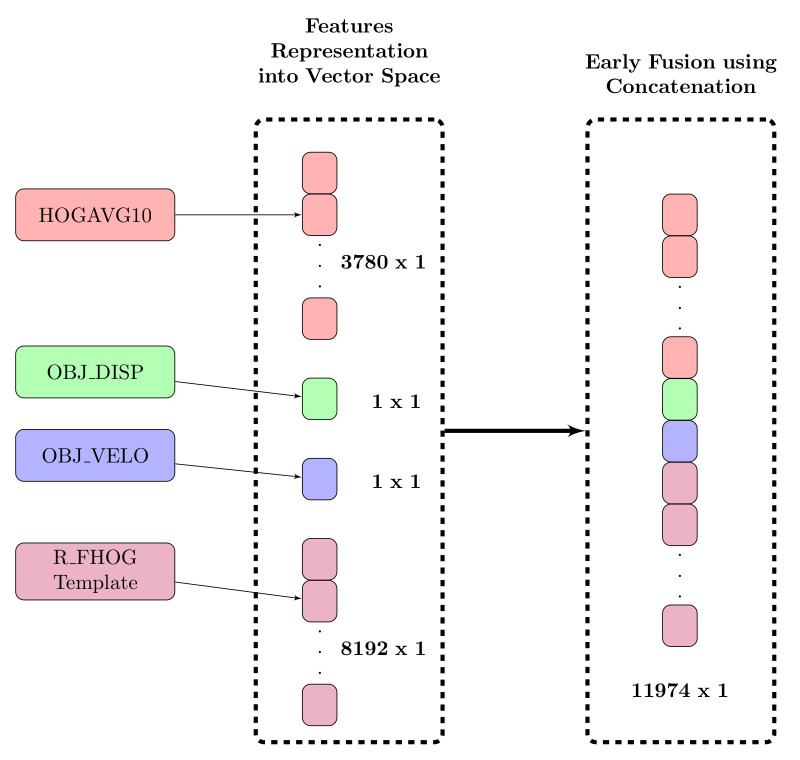
Proposed early fusion technique using concatenation method.

**Figure 10 sensors-20-07299-f010:**
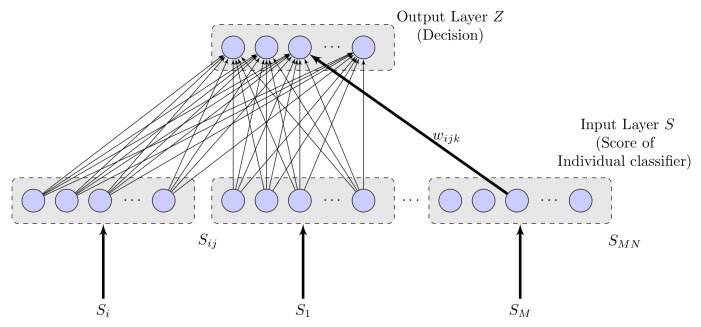
Proposed Late Fusion Technique using Decision Combination Neural Network (DCNN).

**Figure 11 sensors-20-07299-f011:**
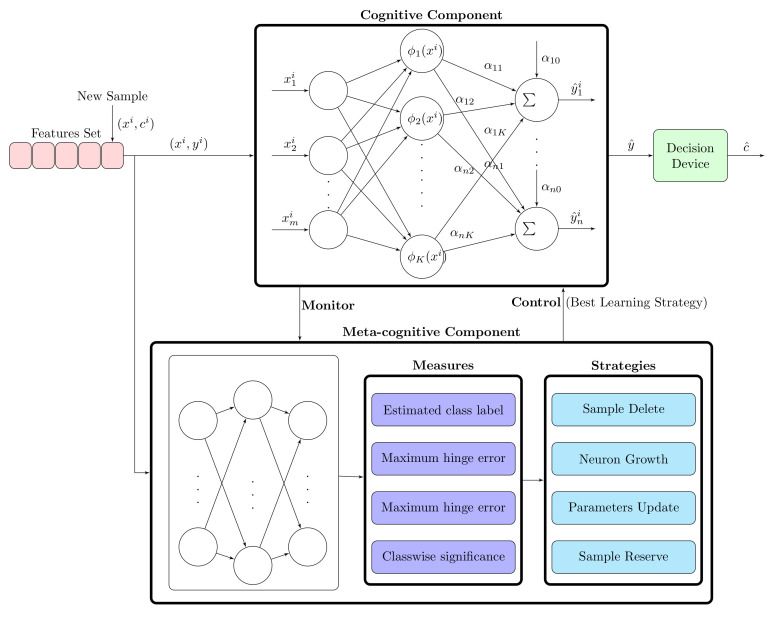
McNN architecture.

**Figure 12 sensors-20-07299-f012:**
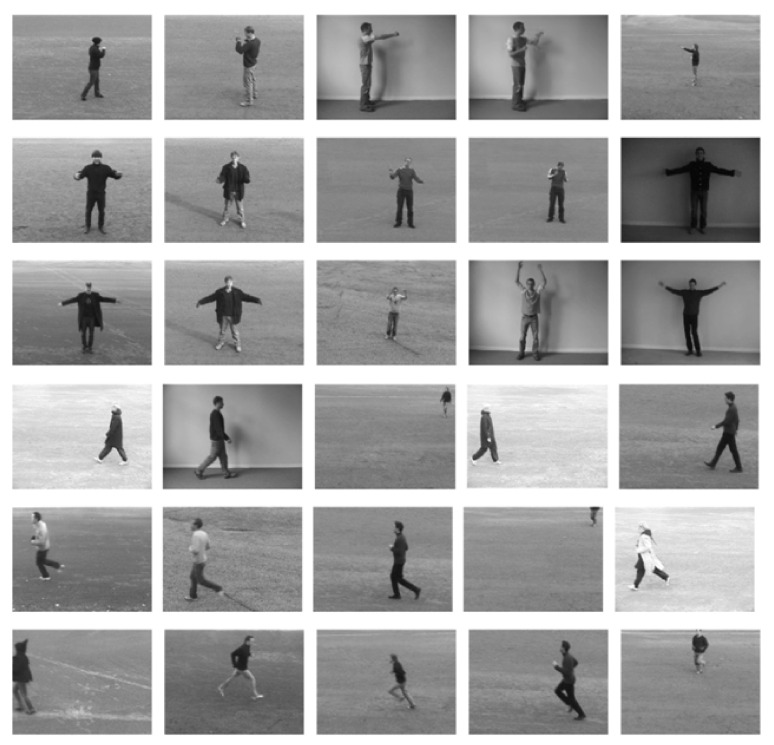
Video sequences from KTH datasets.

**Figure 13 sensors-20-07299-f013:**
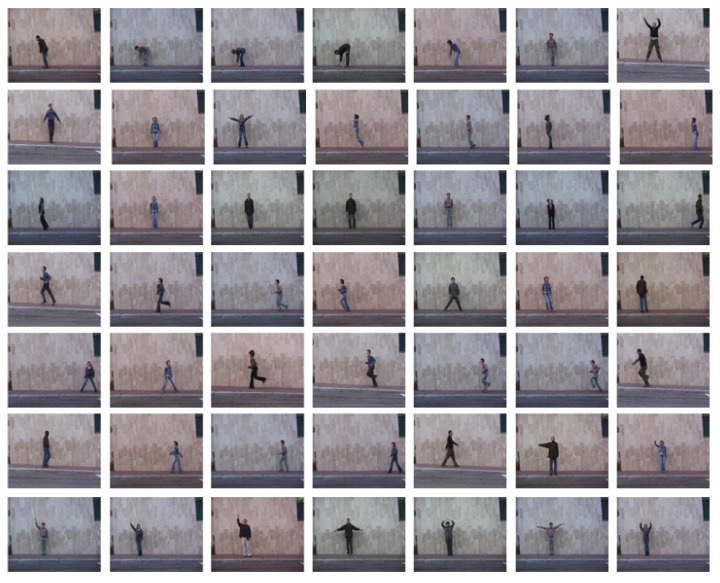
Video sequences from Weizmann datasets.

**Figure 14 sensors-20-07299-f014:**
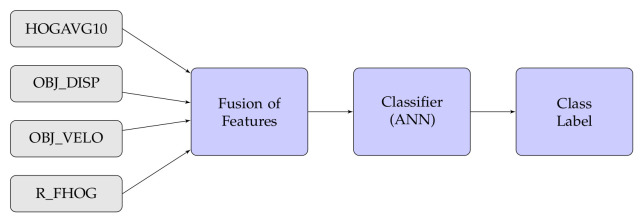
Early fusion with ANN.

**Figure 15 sensors-20-07299-f015:**
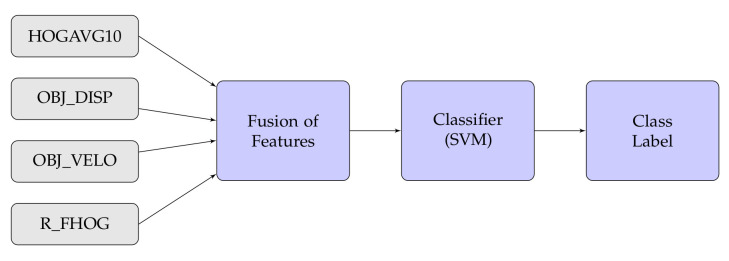
Early fusion with SVM.

**Figure 16 sensors-20-07299-f016:**
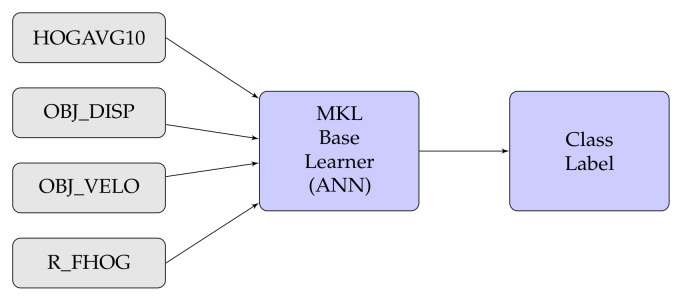
MKL with ANN.

**Figure 17 sensors-20-07299-f017:**
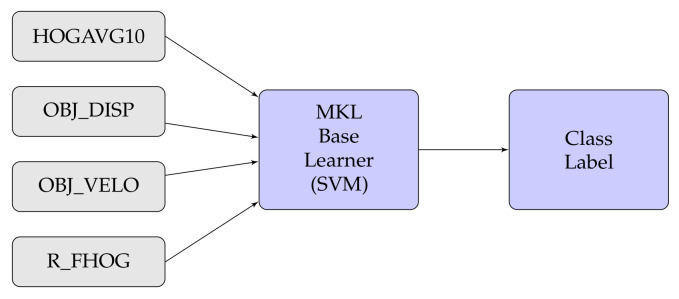
MKL with SVM.

**Figure 18 sensors-20-07299-f018:**
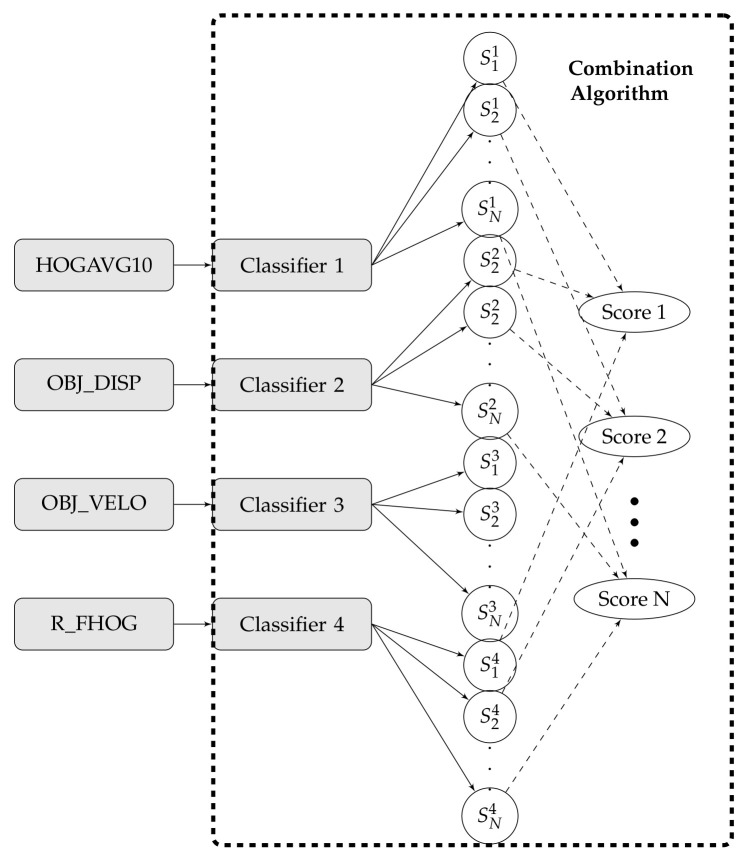
Late fusion.

**Figure 19 sensors-20-07299-f019:**
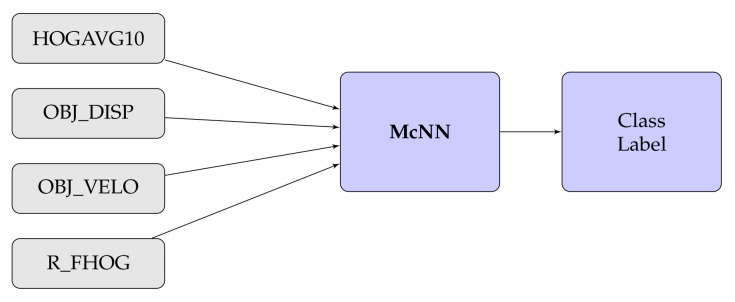
Meta-cognitive neural network.

**Figure 20 sensors-20-07299-f020:**
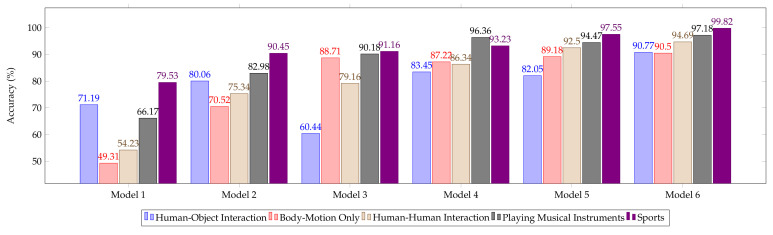
Accuracy comparison of different models on UCF101 dataset action categories.

**Table 1 sensors-20-07299-t001:** Displacement for all classes.

	Non-Zero Displacement Action Type						Zero Displacement Action Type					
**Action**	**Walking**	**Jogging**	**Running**	**Side**	**Skip**	**Jump**	**Boxing**	**Handclapping**	**Handwaving**	**Bend**	**Jack**	**Pjump**
Displacement	15	21.33	64	34	42	51	0	0	0	0	0	0

**Table 2 sensors-20-07299-t002:** Velocity for all classes.

	Non-Zero Velocity Action Type						Zero Velocity Action Type					
**Action**	**Walking**	**Jogging**	**Running**	**Side**	**Skip**	**Jump**	**Boxing**	**Handclapping**	**Handwaving**	**Bend**	**Jack**	**Pjump**
Velocity	37.5	53.325	160	85	105	127.5	0	0	0	0	0	0

**Table 3 sensors-20-07299-t003:** Parameters setting for SVM and their respective levels evaluated in experimentation [36].

Parameters	Levels	Levels
	(Polynomial Kernel)	(Radial Basis)
Degree of Kernel Function (d)	1; 2; 3; 4	-
Gamma in Kernel Function (γ)	-	0.5, 1.0, 1.5, ⋯, 5.0, 10.0
Regularization Parameter (c)	0.5, 1, 5, 10, 100	0.5, 1, 5, 10

**Table 4 sensors-20-07299-t004:** Parameters setting for neural network and their respective levels evaluated in experimentation [36].

Parameters	Level(s)
Hidden Layer Neurons (n)	10, 20, ⋯, 100
Number of Epochs (ep)	1000, 2000, ⋯, 10,000
Momentum Constant values (mc)	0.1, 0.2, ⋯, 0.9
learning Rate Value (lr)	0.1

**Table 5 sensors-20-07299-t005:** Parameter settings for McNN classifier.

Parameter	Strategy-Wise Threshold			
	**Sample Delete**	**Neuron Growth**	**Parameter Update**	**Sample Reverse**
Estimated class label $\hat{c}$				
maximum hinge error *E*		[1.2–1.5]	[0.3–0.8]	
Confidence of classifier $\hat{p}\left(c/x^{i}\right)$	[0.85–0.95]			
Class-wise significance $\psi_{c}$		[0.4–0.8]		

**Table 6 sensors-20-07299-t006:** Confusion matrix for SVM classifier with different kernel functions for KTH dataset.

		Linear SVM						Polynomial SVM						Radial Basis Function SVM					
	**Predicted Class**	**Box**	**Jog**	**Run**	**Walk**	**Wave**	**Clap**	**Box**	**Jog**	**Run**	**Walk**	**Wave**	**Clap**	**Box**	**Jog**	**Run**	**Walk**	**Wave**	**Clap**
**Actual Class**																			
box		**91.50**	0.0	0.0	0.0	5.25	3.25	**96.46**	0.0	0.0	0.0	2.20	1.34	**97.46**	0.0	0.0	0.0	1.68	0.86
jog		0.0	**78.67**	15.53	5.8	0.0	0.0	0.0	**95.71**	3.25	1.04	0.0	0.0	0.0	**97.18**	2.24	0.58	0.0	0.0
run		0.0	14.82	**81.48**	3.7	0.0	0.0	0.0	2.48	**96.04**	1.48	0.0	0.0	0.0	2.16	**97.28**	0.56	0.0	0.0
walk		0.0	7.63	10.0	**82.37**	0.0	0.0	0.0	2.28	2.73	**94.99**	0.0	0.0	0.0	1.60	1.40	**97**	0.0	0.0
wave		4.82	0.0	0.0	0.0	**89.67**	5.51	2.73	0.0	0.0	0.0	**95.23**	2.04	0.58	0.0	0.0	0.0	**97.94**	1.48
clap		5.47	0.0	0.0	0.0	1.36	**93.17**	3.27	0.0	0.0	0.0	1.35	**95.38**	1.29	0.0	0.0	0.0	1.68	**97.03**

**Table 7 sensors-20-07299-t007:** Confusion matrix for Neural Network with different number of Hidden layer for KTH dataset.

		Neural Network with 1 Hidden Layer						Neural Network with 2 Hidden Layer					
	**Predicted Class**	**Box**	**Jog**	**Run**	**Walk**	**Wave**	**Clap**	**Box**	**Jog**	**Run**	**Walk**	**Wave**	**Clap**
**Actual Class**													
box		**97.10**	0.0	0.0	0.0	1.46	1.44	**92.27**	0.0	0.0	0.0	4.98	2.75
jog		0.0	**85.63**	9.09	5.28	0.0	0.0	0.0	**83.40**	11.65	4.95	0.0	0.0
run		0.0	6.49	**88.67**	4.34	0.0	0.0	0.0	10.84	**84.93**	4.23	0.0	0.0
walk		0.0	6.68	7.35	**85.97**	0.0	0.0	0.0	8.41	9.36	**82.23**	0.0	0.0
wave		5.82	0.0	0.0	0.0	**88.37**	5.81	5.71	0.0	0.0	0.0	**86.71**	7.58
clap		8.64	0.0	0.0	0.0	5.26	**86.10**	8.23	0.0	0.0	0.0	4.59	**87.18**

**Table 8 sensors-20-07299-t008:** Confusion matrix for SVM classifier with different kernel functions for Weizmann dataset.

		Linear SVM										Polynomial SVM									
	**Predicted Class**	**Bend**	**jack**	**Jump**	**Sjump**	**Run**	**Side**	**Skip**	**Walk**	**Wave1**	**Wave2**	**Bend**	**Jack**	**Jump**	**Sjump**	**Run**	**Side**	**Skip**	**Walk**	**Wave1**	**Wave2**
**Actual Class**																					
bend		**91.64**	5.46	2.5	0.4	0	0	0	0	0	0	**93.75**	2.45	0	3.1	0.7	0	0	0	0	0
jack		0	**86.23**	3.21	3.54	0	0	0	0	0	0	0	**90.37**	7.25	2.38	0	0	0	0	0	0
jump		0	0	**88.58**	3.73	0	0	2.604	0	0	0	0	3.36	**91.04**	5.1	0	0	0	0	0	0
sjump		0	4.62	11.23	**84.15**	0	0	0	0	0	0	0	0	9.53	**87.34**	3.03	0	0	0	0	0
run		0	0	0	0	**89.20**	2.13	0	3.67	0	0	0	0	0	3.79	**91.39**	0	0	4.32	0	0
side		0	0	0	0	0	**91.97**	1.69	6.34	0	0	0	0	0	0	0	**93.42**	1.93	4.6	0	0
skip		0	0	1.32	2.79	1.96	0	**93.43**	0	0	0	0	0	0	1.72	1.63	2.30	**94.30**	0	0	0
walk		0	0	0	2.9	7.35	0	0	**87.23**	0	0	0	0	0	2.21	7.59	0	0	**90.32**	0	0
wave1		0	0.25	0	0.75	0	0	0	0	**94.76**	4.24	0	0	0	0.64	0	0	0	0	**95.38**	3.93
wave2		0	0	0	0	0	0	0	0	3.75	**96.25**	0	0	0	0	0	0	0	0	2.36	**97.14**

**Table 9 sensors-20-07299-t009:** Confusion matrix for SVM classifier with different kernel functions for Weizmann dataset.

		Radial Basis Function SVM									
	**Predicted Class**	**Bend**	**Jack**	**Jump**	**Sjump**	**Run**	**Side**	**Skip**	**Walk**	**Wave1**	**Wave2**
**Actual Class**											
bend		**100**	0	0	0	0	0	0	0	0	0
jack		0	**99.24**	0.76	0	0	0	0	0	0	0
jump		0	0	**98.37**	1.63	0	0	0	0	0	0
sjump		0	0	1.33	**98.67**	0	0	0	0	0	0
run		0	0	0	1.68	**98.32**	0	0	0	0	0
side		0	0	0	0	0	**99.54**	0	0.46	0	0
skip		0	0	0	0	0	0	**100**	0	0	0
walk		0	0	0	0	0.72	0	0	**99.28**	0	0
wavel		0	0	0	0	0	0	0	0	**100**	0
wave2		0	0	0	0	0	0	0	0	0	**100**

**Table 10 sensors-20-07299-t010:** Confusion matrix for Neural Network with different number of Hidden layer for Weizmann dataset.

		Neural Network with 1 Hidden Layer										Neural Network with 2 Hidden Layer									
	**Predicted Class**	**Bend**	**Jack**	**Jump**	**Sjump**	**Run**	**Side**	**Skip**	**Walk**	**Wave1**	**Wave2**	**Bend**	**Jack**	**Jump**	**Sjump**	**Run**	**Side**	**Skip**	**Walk**	**Wave1**	**Wave2**
**Actual Class**																					
bend		91.42	5.23	3.35	0	0	0	0	0	0	0	93.73	4.16	2.11	0	0	0	0	0	0	0
jack		0	93.21	2.37	4.42	0	0	0	0	0	0	0	94.34	4.51	1.15	0	0	0	0	0	0
jump		0	2.79	91.08	6.13	0	0	0	0	0	0	0	1.68	92.47	5.85	0	0	0	0	0	0
sjump		0	0	3.71	96.29	0	0	0	0	0	0	0	0	2.17	97.23	0	0	0	0	0	0
run		0	0	0	0	89.68	0.54	0	9.78	0	0	0	0	0	0	90.35	1.01	0	8.64	0	0
side		0	0	0	0	0	85.38	8.45	6.17	0	0	0	0	0	0	0	89.78	7.36	2.86	0	0
skip		0	0	0	0	0	9.37	84.65	5.98	0	0	0	0	0	0	0	9.84	86.47	3.67	0	0
walk		0	0	0	0	0	1.72	8.26	90.02	0	0	0	0	0	0	0	1.03	7.38	91.59	0	0
wave1		0	0	0	0	0	0	0	0	86.73	13.27	0	0	0	0	0	0	0	0	89.58	10.42
wave2		0	0	0	0	0	0	0	0	8.66	91.34	0	0	0	0	0	0	0	0	6.11	93.89

**Table 11 sensors-20-07299-t011:** Confusion matrix for RBF SVM for UCF11 dataset.

	Shoot	Bike	Dive	Golf	Ride	Juggle	Swing	Tennis	Jump	Spike	Dog
Shoot	81.3	0	0	0	0	0	13.5	0	0	5.2	0
Bike	0	79.7	0	0	16.7	0	0	0	0	0	3.6
Dive	13.6	0	83	0	0	0	0	0	0	3.4	0
Golf	3.2	0	8.6	85.6	0	1.4	1.2	0	0	0	0
Ride	0	15.9	0	0	78.2	0	0	0	0	1.8	4.1
Juggle	0	0	0	4.8	0	75.8	0	19.4	0	0	0
Swing	0	7.9	0	0	0	0	68.3	0	23.8	0	0
Tennis	0	0	0	14.6	0	9.8	0	71.9	0	3.7	0
Jump	0	0	0	0	5.7	9.9	0	0	84.4	0	0
Spike	0	0	18.5	0	0	0	0	10.9	0	70.6	0
Dog	9.3	0	10.8	0	0	11.2	0	0	0	0	68.7

**Table 12 sensors-20-07299-t012:** Confusion matrix of McNN for KTH dataset.

		Neural Network with 1 Hidden Layer					
	**Predicted Class**	**Box**	**Jog**	**Run**	**Walk**	**Wave**	**Clap**
**Actual Class**							
box		**100**	0.0	0.0	0.0	0	0
jog		0.0	**100**	0	0	0	0
run		0.0	0	**100**	0	0	0
walk		0.0	0	0	**100**	0.0	0.0
wave		0	0.0	0.0	0.0	**100**	0
clap		0	0.0	0.0	0.0	0	**100**

**Table 13 sensors-20-07299-t013:** Confusion matrix of McNN for Weizmann dataset.

	**Predicted Class**	Bend	Jack	Jump	Sjump	Run	Side	Skip	Walk	Wave1	Wave2
**Actual Class**											
bend		**100**	0	0	0	0	0	0	0	0	0
jack		0	**100**	0	0	0	0	0	0	0	0
jump		0	0	**100**		0	0	0	0	0	0
sjump		0	0	0	**100**	0	0	0	0	0	0
run		0	0	0	0	**100**	0	0	0	0	0
side		0	0	0	0	0	**100**	0	0	0	0
skip		0	0	0	0	0	0	**100**	0	0	0
walk		0	0	0	0	0	0	0	**100**	0	0
wavel		0	0	0	0	0	0	0	0	**100**	0
wave2		0	0	0	0	0	0	0	0	0	**100**

**Table 14 sensors-20-07299-t014:** Confusion matrix of McNN for UCF11 dataset.

	Shoot	Bike	Dive	Golf	Ride	Juggle	Swing	Tennis	Jump	Spike	Dog
Shoot	81.3	0	0	0	0	0	13.5	0	0	5.2	0
Bike	0	79.7	0	0	16.7	0	0	0	0	0	3.6
Dive	13.6	0	83	0	0	0	0	0	0	3.4	0
Golf	3.2	0	8.6	85.6	0	1.4	1.2	0	0	0	0
Ride	0	15.9	0	0	78.2	0	0	0	0	1.8	4.1
Juggle	0	0	0	4.8	0	75.8	0	19.4	0	0	0
Swing	0	7.9	0	0	0	0	68.3	0	23.8	0	0
Tennis	0	0	0	14.6	0	9.8	0	71.9	0	3.7	0
Jump	0	0	0	0	5.7	9.9	0	0	84.4	0	0
Spike	0	0	18.5	0	0	0	0	10.9	0	70.6	0
Dog	9.3	0	10.8	0	0	11.2	0	0	0	0	68.7

**Table 15 sensors-20-07299-t015:** State-of-the-Art Comparison of Accuracy of Proposed Approaches for KTH dataset.

Method	Accuracy
Heng et al. 2011 [11]	94.2
Liu et al. 2016 [15]	95.3
Beaudry et al. 2016 [17]	95
Zheng et al. 2018 [20]	94.58
Schuldt et al. 2004 [40]	71.72
Jingen et al. 2009 [42]	91.8
Dollar et al. 2005 [45]	80
Jiang et al. 2006 [46]	84.44
Juan et al. 2008 [47]	83.33
Chuohao et al. 2006 [48]	86
Ke et al. 2007 [49]	81
Kim et al. 2007 [50]	95.33
Jhuang et al. 2007 [51]	91.6
Laptev et al. 2008 [52]	91.83
Rapantzikos et al. 2009 [53]	88.30
Bregonzio et al. 2009 [54]	93.17
Klaser et al. 2008 [55]	91.4
Fathi et al. 2008 [56]	90.50
Le et al. 2011 [57]	93.9
Kovashka et al. 2010 [58]	94.53
Yeffet et al. 2009 [59]	90.1
Wang et al. 2013 [60]	95.3
**Early Fusion using ANN**	**84.12**
**Early Fusion using SVM**	**92.32**
**MKL with ANN**	**93.03**
**MKL with SVM**	**95.85**
**Late Fusion using DCNN**	**96.19**
**Late Fusion using**	**98.98**
**Fuzzy Integral**	
**McNN**	**100**

**Table 16 sensors-20-07299-t016:** State-of-the-Art Comparison of Accuracy of Proposed Approaches for Weizmann dataset .

Method	Accuracy
Liu et al. 2016 [15]	100
Lena et al. 2009 [41]	88.2
Bregonzio et al. 2009 [54]	96.66
Klaser et al. 2008 [55]	84.3
Grundmann et al. 2008 [61]	96.39
Weinland et al. 2008 [62]	93.33
Nguyen et al. 2011 [63]	87.7
Ballan et al. 2009 [64]	92.41
Yang et al. 2009 [65]	97.2
Chen et al. 2009 [66]	100
Vezzani et al. 2010 [67]	86.7
Dhillon et al. 2009 [68]	88.5
Lin et al. 2009 [69]	100
Natarajan et al. 2010 [70]	99.5
Yan et al. 2009 [71]	99.4
**Early Fusion using ANN**	**91.943**
**Early Fusion using SVM**	**94.34**
**MKL with ANN**	**92.09**
**MKL with SVM**	**93.89**
**Late Fusion using DCNN**	**95.25**
**Late Fusion using**	**97.97**
**Fuzzy Integral**	
**McNN**	**100**

**Table 17 sensors-20-07299-t017:** State-of-the-Art Comparison of Accuracy of Proposed Approaches for UCF11 dataset .

Method	Accuracy
Wang et al. 2011 [11]	84.2
Liu et al. 2009 [72]	71.2
Ikizler et al. 2010 [73]	75.2
Mota et al. 2013 [74]	72.7
Sad et al. 2013 [75]	72.6
Wang et al. 2013 [60]	89.9
Figueiredo et al. 2014 [76]	59.5
Hasan et al. 2014 [77]	54.5
Kihl et al. 2014 [78]	86.0
Maia et al. 2015 [79]	64.0
Patel et al. 2016 [80]	89.43
**Early Fusion using ANN**	**69.96**
**Early Fusion using SVM**	**74.05**
**MKL with ANN**	**75.07**
**MKL with SVM**	**78.38**
**Late Fusion using DCNN**	**79.88**
**Late Fusion using**	**82.12**
**Fuzzy Integral**	
**McNN**	**89.93**

**Table 18 sensors-20-07299-t018:** State-of-the-Art Comparison of Accuracy of Proposed Approaches for HMDB-51 dataset .

Method	Accuracy
Liu et al. 2009 [72]	71.2
Kuehne et al. 2011 [43]	23.0
Kliper et al. 2012 [81]	29.2
Wang et al. 2013 [60]	46.6
Wang et al. 2013 [12]	57.2
Can et al. 2013 [82]	39.0
Ni et al. 2015 [13]	66.7
Lan et al. 2015 [14]	65.1
Liu et al. 2016 [15]	48.4
Hongyang et al. 2016 [16]	60.3
Beaudry et al. 2016 [17]	49.6
Liu et al. 2016 [83]	58.1
Ahsan et al. 2018 [19]	28.5
Lin et al. 2018 [21]	63.0
Lan et al. 2017 [84]	75
Zhu et al. 2018 [85]	74.8
Zhu et al. 2018 [84]	78.7
Carreira et al. 2018 [85]	80.2
**Early Fusion using ANN**	**44.68**
**Early Fusion using SVM**	**49.32**
**MKL with ANN**	**52.43**
**MKL with SVM**	**54.19**
**Late Fusion using DCNN**	**55.02**
**Late Fusion using**	**55.89**
**Fuzzy Integral**	
**McNN**	**67.03**

**Table 19 sensors-20-07299-t019:** State-of-the-Art Comparison of Accuracy of Proposed Approaches for UCF101 dataset .

Method	Accuracy
Wang et al. 2013 [12]	86
Simonyan et al. 2014 [86]	88
Karpathy et al. 2014 [87]	65.4
Donahue et al. 2015 [88]	82.66
Sun et al. 2015 [89]	88.1
Lan et al. 2015 [14]	89.1
Feichtenhofer et al. 2016 [90]	92.5
Zhang et al. 2016 [91]	86.4
Cherian et al. 2017 [92]	94.6
Seo et al. 2017 [93]	85.74
Shi et al. 2017 [94]	92.2
Wang et al. 2017 [95]	91.32
Zheng et al. 2018 [20]	95.1
Ahsan et al. 2018 [19]	67.1
Zheng et al. 2018 [20]	95.1
Lin et al. 2018 [21]	91.5
Lan et al. 2017 [84]	95.3
Zhu et al. 2018 [85]	95.8
Zhu et al. 2018 [84]	97.1
Carreira et al. 2018 [85]	97.9
**Early Fusion using ANN**	**64.23**
**Early Fusion using SVM**	**79.87**
**MKL with ANN**	**81.93**
**MKL with SVM**	**89.32**
**Late Fusion using DCNN**	**91.87**
**Late Fusion using**	**93.15**
**Fuzzy Integral**	
**McNN**	**94.59**

**Table 20 sensors-20-07299-t020:** Classification accuracy against the state-of-the-art on HMDB51 and UCF101 datasets averaged over three splits with CNN architectures.

Method	UCF101	HMDB51
Two Stream CNN [86]	88	59.4
Slow Fusion CNN [87]	65.4	-
EMV+RGB-CNN [91]	86.4	-
Spatio-temporal CNN [89]	88.1	59.1
Very Deep Two Stream Fusion [90]	93.5	69.2
Generalized Rank Pooling [92]	93.5	72.0
Frame Skipping + Trajectories Rejection [93]	85.74	58.91
Three-stream sDTD [94]	92.2	65.2
Order Pooling [95] (Dyn. Flow+RGB+(S)Op.Flow+IDT-FV)	91.32	67.35
Deep Feature [84]	95.3	75
End-to-End video [85]	95.8	74.8
Two Stream CNN [96]	97.1	78.7
Kinetics [97]	97.9	80.2
Proposed Approach (McNN)	94.59	67.03

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
