# Peer review of "Histogram of Oriented Gradient-Based Fusion of Features for Human Action Recognition in Action Video Sequences"

_sensors, 2020, doi:10.3390/s20247299_

Round 1

Reviewer 1 Report

The paper introduces a framework to address human action recognition on constrained video sequences that involve moving background and multiple actions in a single video sequence.

The framework relies on the background removing approach already published by the authors in 2014.

The procedure for extracting feature descriptors from a segmented object is then performed by combining HOG, object displacement, object velocity, and Regional Features from Fourier HOG (proposed in 2014 by Liu et al). The task Feature Fusion is performed using basic techniques such as concatenating features and finally a Decision Combination of Neural Network (DCNN)  (dating back to 1995 is used for classification).

Unfortunately, I can't see any relevant scientific contributions from the proposed framework. Authors should better explain where the reader should look to find the advance in the knowledge. 

The comparison with other approaches is convincing even if performance is quite similar to the most effective past approaches.

The authors should clarify!

Minor concerns:

There are some flaws in the paper format (some tables go out of margins)  and figure 28 is missing.

Introduction miss some portion of scientific contents about background subtraction (especially advanced approaches using statistical and semantic reasonings and recent works using end to end CNN). Examples of missing references concerning background subtraction are:

[1] Spagnolo, P., D’Orazio, T., Leo, M., & Distante, A. (2005, September). Advances in background updating and shadow removing for motion detection algorithms. In International Conference on Computer Analysis of Images and Patterns (pp. 398-406). Springer, Berlin, Heidelberg.

[2] Mondéjar-Guerra, V. M., Rouco, J., Novo, J., & Ortega, M. (2019, September). An end-to-end deep learning approach for simultaneous background modeling and subtraction. In BMVC (p. 266).

Also, the discussion about HAR misses relevant papers such as

[3] Shi, L., Zhang, Y., Cheng, J., & Lu, H. (2019). Skeleton-based action recognition with directed graph neural networks. In Proceedings of the IEEE Conference on Computer Vision and Pattern Recognition (pp. 7912-7921).

Author Response

The authors would like to thank the reviewers for their valuable suggestions which helped in improving the quality of this paper.

Reviewer 2 Report

1) Authors should comment relation of proposed method to deep learning approaches from literature. Why design with handcrafted features is chosen instead of deep learning approach?

2) Line 45: "Firstly, for moving 46 object detection, we use a novel technique by incorporating the human visual attention model [11] making it background independent."

Line 125: "However, an algorithm for moving object detection without any background modeling was presented in [22]."

It is stated that object detection method represents novelty, but it is obviously taken from literature. Another question is whether it is method presented in [11] or method from [22].  

3) Line 126, equation 3.

How is distance d defined? As intensity difference?

4) Line 151: „centroid (or center of mass) of the silhouette  corresponding to the object is calculated by taking the (arithmetic) mean of the pixels“

Mean of pixel locations?

5) Line 236, equation (19).

Is equal sign missing?

6) Lines 252-253.

Unclear sentence?

7) Line 405.

Figure 28 is missing.

8) For HMDB51 and UCF101 datasets performance comparison presented in tables 20, 21 and 22  should include more methods (see [1], table 2)

9) Lines 421-423.

Feature descriptor is designed with the focus on KTH and Weizmann datasets what results with high accuracy on these datasets. On more complex datasets accuracy is lower, but still comparable with state of the art methods. Authors should explain this difference in more details, that is why proposed descriptor is less appropriate for certain datasets. Also, it should be commented why result for HMDB51 dataset is significantly lower than in case of UCF101 dataset.

10) Line 434:“ …our McNN with proposed features outperforms the best result.“

Comparison should include recent methods with accuracy higher than proposed method (see [1], table 2).

[1] Zhang, H. B., Zhang, Y. X., Zhong, B., Lei, Q., Yang, L., Du, J. X., & Chen, D. S. (2019). A comprehensive survey of vision-based human action recognition methods. Sensors, 19(5), 1005.

Author Response

(The authors gave the same response as above.)

Round 2

Reviewer 1 Report

The updated version partially solved my concerns. Not having considered deep learning is definitely a limitation and the authors did not provide an adequate explanation in the paper to this choice.
Readers need it in order to place the paper in the literature.
Anyway, the manuscript deserves a chance of publication because the proposed pipeline is interesting.

Author Response

Query:-1

The updated version partially solved my concerns. Not having considered deep learning is definitely a limitation and the authors did not provide an adequate explanation in the paper to this choice.
Readers need it in order to place the paper in the literature.
Anyway, the manuscript deserves a chance of publication because the proposed pipeline is interesting.

Response:-

The concern from the reviewer is correct that we have not included the Deep Learning Techniques over here for Human Action Recognition.

The major Purpose of the manuscript is to generate Robust Feature Descriptor Preparation. In addition to that, we have used various fusion techniques for Feature as well as for Classifiers.

In Future Scope, we are considering the Deep Learning Technique for solving the Human Action Recognition Problem

Reviewer 2 Report

Query 3

Add equation for intensity difference to text.

Query 4:

Add „mean of pixel locations“ to text.

Query 9:

Explanation from response should be added to text.

Query 10:

Sentence „ For UCF101 dataset, we find that our McNN with proposed features outperforms the best result“ should be changed because proposed method has lower accuracy in comparison to added methods.

Author Response

Query 3

Add equation for intensity difference to text.

Response:-Page number :-5, equation number :-3 and line number 120

Query 4:

Add „mean of pixel locations“ to text.

Response:- Page number :-6, equation number :-6 and line number 155

Query 9:

Explanation from response should be added to text.

Response:- Page number :-25 and line number 550

Query 10:

Sentence „ For UCF101 dataset, we find that our McNN with proposed features outperforms the best result“ should be changed because proposed method has lower accuracy in comparison to added methods.

Response:- Page number :-25 and line number 555